# Intramolecular crossover from unconventional diamagnetism to paramagnetism of palladium ions probed by soft X-ray magnetic circular dichroism

Alevtina Smekhova 1,5, Detlef Schmitz[2], Natalya V. Izarova[1], Maria Stuckart[1,3,6], S. Fatemeh Shams 1, Konrad Siemensmeyer[2], Frank M. F. de Groot[4], Paul Kögerler 1,3 & Carolin Schmitz-Antoniak 1✉

The case of palladium(II) ions in molecular polyoxopalladates highlights the importance of accounting not only for nearest neighbour atoms or ions in order to understand, model or predict magnetic characteristics. Here, using site-specific soft X-ray magnetic circular dichroism (XMCD), the effects of different bond lengths, delocalization of 4d electrons, and 4d spin-orbit coupling on the electronic and magnetic properties are investigated and three different states identified: Conventional diamagnetism in a square-planar $O_4$ coordination environment, paramagnetism caused by four additional out-of-plane oxygen anions, and an unusual diamagnetic state in the diamagnetic/paramagnetic crossover region modified by significant mixing of states and facilitated by the substantial 4d spin-orbit coupling. The two diamagnetic states can be distinguished by characteristic XMCD fine structures, thereby overcoming the common limitation of XMCD to ferro-/ferrimagnetic and paramagnetic materials in external magnetic fields. The qualitative interpretation of the results is corroborated by simulations based on charge transfer multiplet calculations and density functional theory results.

[1] Peter-Grünberg-Institut (PGI-6), Forschungszentrum Jülich, 52425 Jülich, Germany. [2] Helmholtz-Zentrum Berlin für Materialien und Energie, Albert-Einstein-Str. 15, 12489 Berlin, Germany. [3] Institut für Anorganische Chemie, RWTH Aachen University, Landoltweg 1, 52074 Aachen, Germany. [4] Inorganic Chemistry and Catalysis Group, Debye Institute for Nanomaterials Science, Utrecht University, Universiteitsweg 99, 3584 CG Utrecht, The Netherlands. [5] Present address: Helmholtz-Zentrum Berlin für Materialien und Energie, Albert-Einstein-Str. 15, 12489 Berlin, Germany. [6] Present address: Department Chemie- und Bioingenieurwesen, Lehrstuhl für Chemische Reaktionstechnik (CRT), Friedrich-Alexander Universität Erlangen-Nürnberg, Egerlandstr. 3, 91058 Erlangen, Germany. ✉email: c.schmitz-antoniak@fz-juelich.de

Polyoxopalladates represent a new class of nanoscale metal oxide materials with well-defined molecular structures and a wide range of derivatives derived from set of archetypal cluster architectures. The first example of this class, the polyanion $[Pd^{II}_{13}O_2(OH)_6(As^VO_4)_8]^{8-}$, denoted PdPd$_{12}$As$_8$ (4) in this work, was discovered in 2008[1]. It is composed of a central palladium(II) ion surrounded by a distorted cubic shell of eight oxygen ions, which is in turn encapsulated in the cuboctahedral cavity of 12 palladium(II) ions in square-planar oxygen coordination environments ({Pd$_{12}$}) and capped by eight arsenate (AsO$_4^{3-}$) heterogroups. Although surrounded by eight oxygen anions, the symmetry of the central palladium(II) ion was initially reported as square-planar[1]—based on the corresponding shortest Pd–O distances—and is sketched in Fig. 1c.

Replacing the central palladium by an iron ion while retaining the symmetry and number of palladium(II) ions in the outer shell allows to discern the contributions from the central and outer shell palladium to element-specific X-ray absorption spectra. The $[Fe^{III}O_8Pd^{II}_{12}(PhAs^VO_3)_8]^5$ polyanion[2], here denoted FePd$_{12}$(PhAs)$_8$ (3), contains an iron(III) ion in its centre, coordinated by eight nearest oxygen ions, and comprises eight phenyl arsonate heterogroups instead of arsenates (Fig. 1b). Moreover, all Fe–O bond lengths in FePd$_{12}$(PhAs)$_8$ are equal and the {Pd$_{12}$} shell in FePd$_{12}$(PhAs)$_8$ is similar but less distorted than in PdPd$_{12}$As$_8$. Significantly smaller Pd–O bond lengths in the {Pd$_{12}$} shell result when utilizing phosphate groups as capping groups[3] in $[Fe^{III}O_8Pd^{II}_{12}(P^VO_4)_8]^{13-}$, denoted FePd$_{12}$P$_8$ (2), and $[Co^{II}O_8Pd^{II}_{12}(P^VO_4)_8]^{14-}$, denoted CoPd$_{12}$P$_8$ (1). The relevant bond lengths, their variance as well as location and formal coordination symmetry of palladium(II) ions of the four different samples are summarized in Table 1.

From coordination symmetry, Co$^{2+}$ (3$d^7$) and Fe$^{3+}$ (3$d^5$) ions in CoPd$_{12}$P$_8$, FePd$_{12}$P$_8$, and FePd$_{12}$(PhAs)$_8$ are paramagnetic with a high-spin state, because the crystal field splitting in eightfold (cubic) symmetry is usually small in comparison to the spin pairing energy. It has already been evidenced experimentally[4] that the magnetism of Fe and Co central ions in polyoxopalladates can be well described within this simple model. The Pd$^{2+}$ (4$d^8$) ions of the {Pd$_{12}$} shell in square-planar symmetry are expected to be diamagnetic since the orbital in the coordination plane pointing to the oxygen anions is usually energetically unfavoured and remains unoccupied. Consequently, conventional magnetometry

measurements are dominated by the paramagnetic response of the central Fe or Co ion[4].

The square-planar coordination of the central Pd$^{2+}$ (4$d^8$) ion in PdPd$_{12}$As$_8$ should again lead to a diamagnetic configuration response as explained above for the Pd$^{2+}$ ions in the shell. However, here we present an unexpected sizeable paramagnetic moment which is reproducibly observed also for PdPd$_{12}$As$_8$ by magnetometry and motivated further investigations.

Measurements of the X-ray absorption near-edge structure (XANES) and its associated X-ray magnetic circular dichroism (XMCD) have the advantage to be element-specific and even site-specific in contrast to conventional magnetometry. Hence, XANES and XMCD spectra recorded at the Pd $M_{3,2}$ absorption edges are used to explore electronic and magnetic properties of palladium(II) ions in the central or shell positions. Thereby, the consequences of Pd–O bond lengths, crystal symmetry and coordination on the magnetic properties of palladium(II) ions are investigated. Simulations of XANES and XMCD using the CTM4XAS program package[5] support our results by clearly disentangling the site-specific contributions to the experimental spectra. In addition, density functional theory (DFT) calculations of the ground state properties are performed to further substantiate our conclusions.

## Results

**Magnetometry.** Conventional magnetometry was carried out in magnetic fields up to 5 T and temperatures down to 2 K for samples (1)–(4). The data are presented in Fig. 2. As expected, the magnetic moments as a function of magnetic field divided by temperature follow Brillouin functions, thus indicating the dominating Langevin paramagnetism of the central Co$^{2+}$ or Fe$^{3+}$ ions in samples (1)–(3). Surprisingly, sample (4) that contains a Pd$^{2+}$ central ion in formal square-planar coordination symmetry, shows a clear paramagnetic response as well.

**Ground state properties from DFT.** DFT calculations were performed using the B3LYP hybrid functional in order to identify and characterize possible paramagnetism of PdPd$_{12}$As$_8$. Starting from different structural inputs, the system was allowed for geometric relaxation of either a diamagnetic ($S = 0$) or a paramagnetic ($S = 1$) state of PdPd$_{12}$As$_8$, more precisely of a $[Pd^{II}_{13}As^V_8O_{40}]^{14-}$ assembly not including protonation. In fact,

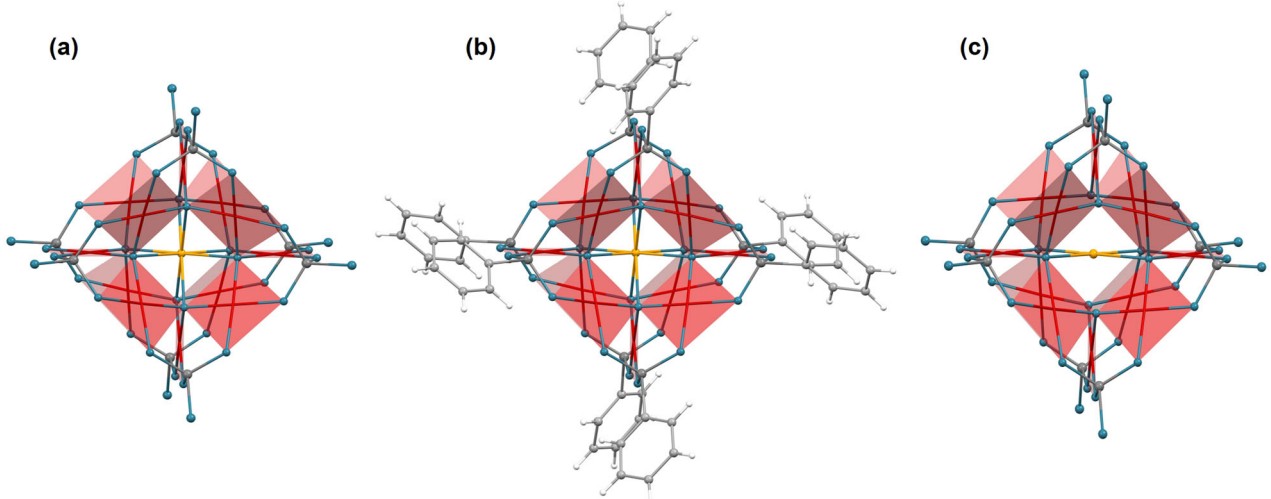

**Fig. 1 Molecular geometries.** Ball-and-stick representation of **a** CoPd$_{12}$P$_8$ or FePd$_{12}$P$_8$, **b** FePd$_{12}$(PhAs)$_8$ and **c** PdPd$_{12}$As$_8$. The 12 square-planar PdO$_4$ groups in the shell are highlighted as transparent red planes. The central ions (i.e. Co$^{2+}$ or Fe$^{3+}$ in **a**, Fe$^{3+}$ in **b**, and Pd$^{2+}$ in **c**, respectively) are sketched in orange, oxygen in blue, P or As in dark grey, C in light grey, and H (phenyl groups) in white.

**Table 1 Mean Pd–O bond lengths.**

| | Sample | $d_{mean}^{Pd-O}$ (Å) | $\Delta_{max}$ (%) | Pd location | Formal coordination | Ref. |
|---|---|---|---|---|---|---|
| (1) | $CoPd_{12}P_8$ | 1.994 | 5.3 | Shell | Square-planar | 3 |
| (2) | $FePd_{12}P_8$ | 2.000 | 2.9 | Shell | Square-planar | 3 |
| (3) | $FePd_{12}(PhAs)_8$ | 2.009 | 4.1 | Shell | Square-planar | 2 |
| (4) | $PdPd_{12}As_8$ | 2.020 | 6.9 | Shell + centre | Square-planar | 1 |

Mean Pd–O bond lengths $d_{mean}^{Pd-O}$, maximum variations $\Delta_{max}$ of bond lengths which are much larger than the experimental uncertainties, as well as location and formal coordination symmetry of palladium (II) ions in palladate samples investigated in this work.

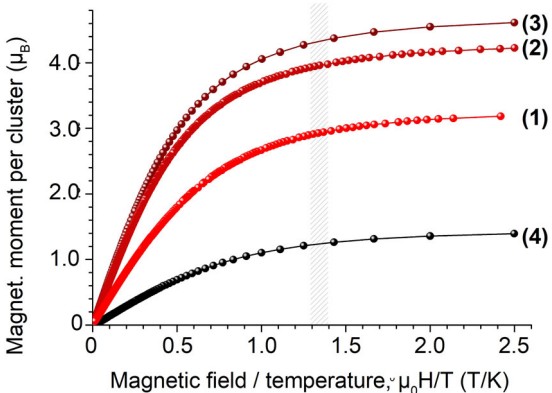

**Fig. 2 Magnetometry.** Conventional magnetometry in magnetic fields up to 5 T and temperatures down to 2 K of $CoPd_{12}P_8$ (1), $FePd_{12}P_8$ (2), $FePd_{12}(PhAs)_8$ (3), and $PdPd_{12}As_8$ (4). A typical paramagnetic response is visible for all samples. The shaded area marks the $\mu_0 H/T$ region in which XMCD analyses of Pd magnetic moments were done.

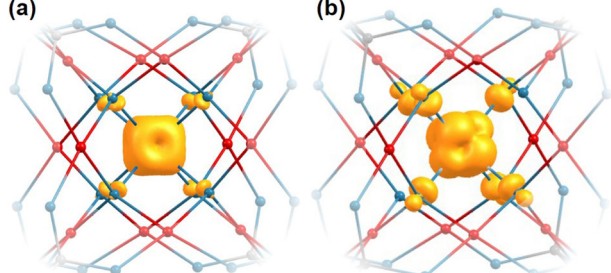

**Fig. 3 Spin densities.** Spin densities calculated for relaxed geometrical structures of **a** $CoPd_{12}P_8$ and **b** $PdPd_{12}As_8$. The yellow surfaces correspond to isovalues of $0.01 r_B^{-3}$. For clarity, the crystal structure close to the central ion is sketched in addition (Pd in red, O in blue, P or As in grey).

the lowest energy was found for the paramagnetic case with usual Pd–O bond lengths (2.03–2.17 Å) and square-planar coordination of Pd ions in the shell, while the coordination symmetry of the central Pd ion is highly distorted. All computed distances were slightly larger than the corresponding experimentally observed atomic distances. This discrepancy between gas-phase calculations and crystallized material in the experiment is already known[2] and can be reduced by introducing a continuous solvation model. However, geometric optimization including the conductor-like polarizable continuum model (CPCM) lead only to reduced bond lengths but retained the symmetry of the lowest-energy structure.

The electronic charge was estimated using the simple Mulliken population analysis (including CPCM), which is the most common method in computational chemistry despite its obvious limitations like, e.g. the dependence of atomic charges on the choice of the basis set. For our X-ray absorption analysis, only the number of unoccupied $d$ states $n_h^d$ is relevant. The 12 $Pd^{2+}$ ions in the shell have similar numbers of unoccupied $d$ states, i.e. on average $n_h^d = 1.46$, 1.47, 1.46, and 1.46 for $CoPd_{12}P_8$ (1), $FePd_{12}P_8$ (2), $FePd_{12}(PhAs)_8$ (3), and $PdPd_{12}As_8$ (4), respectively. For the central Pd ion in sample (4) the number of unoccupied $d$ states is slightly larger, $n_h^d = 1.70$, i.e. it carries less $d$ charge. Since the basis set was the same for all palladate molecules, the qualitative difference between the $Pd^{2+}$ ions at different sites is reliable. Moreover, the absolute values seem to be reasonable since hybridization effects can reduce the effective number of unoccupied states of palladium(II) ions from its free ion value $n_h^d = 2$. This effect is larger for the palladium(II) ions in the shell according to the shorter averaged Pd–O bond lengths. In Fig. 3 we illustrate the spin densities calculated for the relaxed

structures of $PdPd_{12}As_8$ and $CoPd_{12}P_8$. In both cases, the yellow surface corresponds to an isovalue of $0.01 \, r_B^{-3}$ ($r_B$: Bohr radius). Animated rotations of the isosurfaces are presented in Supplementary Movies 1 and 2 for $CoPd_{12}P_8$ and $PdPd_{12}As_8$, respectively. It can be easily seen that the uncompensated spins responsible for the paramagnetism are related to the central ions and not to the palladium in the shell. For the chosen isovalue, the spin imbalance at the neighbouring oxygen anions due to hybridization effects is visible as well. In the case of $CoPd_{12}P_8$, all eight oxygen anions around the Co central ion show this feature in agreement to the reported cubic symmetry of the central ion[3,4]. For the case of the Pd central ion, two main differences can be observed: The isosurfaces around the oxygen anions are larger and they are only present around six of eight neighbouring oxygen anions. The larger isosurfaces at oxygen anions are related to the more delocalized character of the $4d$ electrons compared to $3d$ electrons. However, for two of eight neighbouring oxygen anions, the Pd–O distance seems to be too large (around 2.7 Å) to participate in a classical bond. Thus, as a result from DFT the paramagnetic response of $PdPd_{12}As_8$ to an external magnetic field (Fig. 2) is related to the paramagnetic central ion in a distorted six-fold coordination.

**X-ray absorption spectroscopy.** The averaged XANES and XMCD spectra recorded at the Pd $M_{3,2}$ absorption edges at low temperatures of 4.3–4.6 K in static external magnetic fields of ±6 T are shown in Fig. 4. These absorption edges correspond to electron transitions from the spin–orbit split $3p_{3/2}$ ($M_3$) and $3p_{1/2}$ ($M_2$) states into unoccupied $4d$ states. Due to an overlap with the oxygen K absorption edge ($1s$ to $2p$ transitions), the pure $M_3$ edge (around 532 eV) could not be well resolved for the palladate samples, i.e. further analysis is needed to determine which ones of the spectral features A–E refer to the Pd $M_3$ or O K edge. By comparison with reference spectra measured at the O K edge of $CoFe_2O_4$ nanoparticles that do not contain any palladium (Supplementary Fig. 2 and Supplementary Note 1), one can already

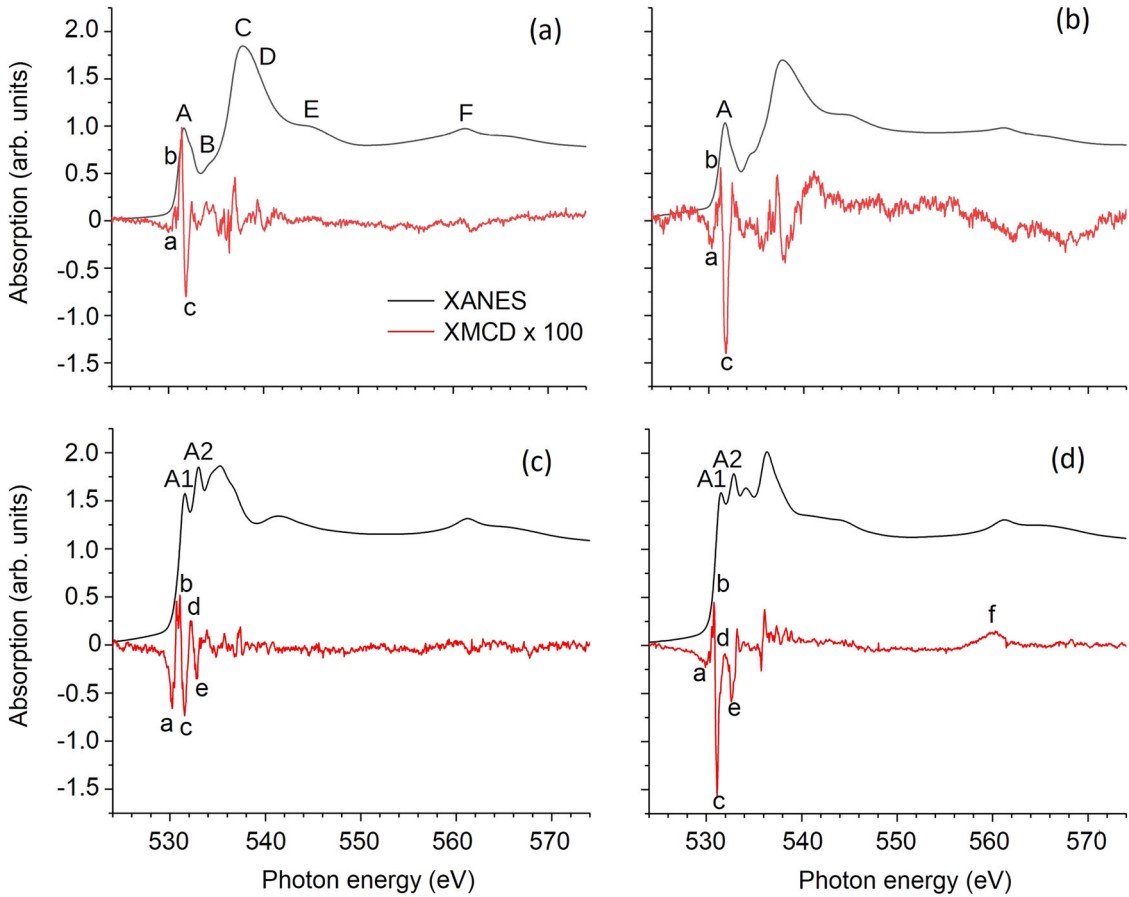

**Fig. 4 Experimental X-ray absorption spectra.** Normalized experimental X-ray absorption near-edge structure (XANES, black lines) and X-ray magnetic circular dichroism (XMCD, red lines) at the Pd $M_{3,2}$ and O K absorption edges of $CoPd_{12}P_8$ **a**, $FePd_{12}P_8$ **b**, $FePd_{12}(PhAs)_8$ **c** and $PdPd_{12}As_8$ **d** in static magnetic fields of ±6 T at a temperature of 4.3–4.6 K. XMCD spectra were multiplied by a factor of 100.

conclude that the features C–E are related to the O K edge. By comparison with a reference spectrum of $[PdCl_4]^{2-}$, peaks A, A1, and A2 can be related to the Pd $M_3$ absorption edge. However, the peaks are expected to be superimposed by oxygen pre-edge features due to hybridization effects. The sharper signature around 560 eV (F) is caused by the Pd $M_2$ absorption, while the oxygen contributes only as a smooth background in this energy range.

Since the energy position and shape of O K edge absorption peaks are sensitive to the type, valence, and coordination of neighbouring atoms, the most striking differences in the XANES spectra of the four samples can already be explained by the contributions of different capping groups and the hybridization with either $Co^{2+}$, $Fe^{3+}$, or $Pd^{2+}$ central ions.

To gain further insight into the fine structures, the experimental XANES spectra around the Pd $M_3$ absorption edge are compared in more detail. In Fig. 5a, b all the spectra—including the one of the oxygen-free $[PdCl_4]^{2-}$ reference—are shown after subtracting two step-like functions accounting for electron transitions into higher unoccupied states than Pd 4d or O 2p, respectively. The palladate absorption edges are shifted to slightly lower energies with respect to the reference sample. This can be related to a smaller effective charge that has already been found for the Pd ions in palladate samples by DFT[6].

For stronger ligands (as in $[PdCl_4]^{2-}$) or shorter Pd–O bond lengths (for $CoPd_{12}P_8$, $FePd_{12}P_8$), there is only one peak visible that can be related to palladium(II) ions while for larger bond lengths (in $FePd_{12}(PhAs)_8$, $PdPd_{12}As_8$) a double-peak feature is visible. In fact, the transition from a single peak to a double-peak

XANES can be simulated, e.g. by slightly decreasing the crystal field as shown in Fig. 5c. The transition can also be triggered by changing the d–d Coulomb repulsion or a modified charge transfer as will be discussed later.

The XMCD spectra of $CoPd_{12}P_8$ (1) and $FePd_{12}P_8$ (2) show a clear negative–positive–negative spectral feature (a–c) around 531 eV (close to XANES peak A). The XMCD spectrum of $FePd_{12}(PhAs)_8$ (3) exhibits a rich fine structure at the energy around the Pd $M_3$ (and O K) absorption edge (denoted a–e) while only a weak signal is obtained at the Pd $M_2$ absorption edge. The XMCD spectrum of $PdPd_{12}As_8$ (4) shows a rich fine structure at the Pd $M_3$ (and O K) absorption edge as well, but with specific differences: The decreasing XMCD intensity before the absorption edge is slightly pronounced, while the feature denoted a is much smaller compared to the case of $FePd_{12}(PhAs)_8$. Moreover, the difference in the peak heights between b and c is larger while d and e largely maintain their difference in amplitude but are both shifted downwards. At a first glance, the changes may indicate an additional peak with negative amplitude and quite broad line width compared to the other spectral features. Another striking difference is the appearance of a distinct XMCD signal at the Pd $M_2$ absorption edge (denoted f) solely for the sample $PdPd_{12}As_8$ (4).

**Fitting of experimental data**. XANES and XMCD spectra for the palladium(II) contributions were simulated by using the CTM4XAS program package[5] and fitted to the experimental data by introducing a small energy shift and a constant scaling factor.

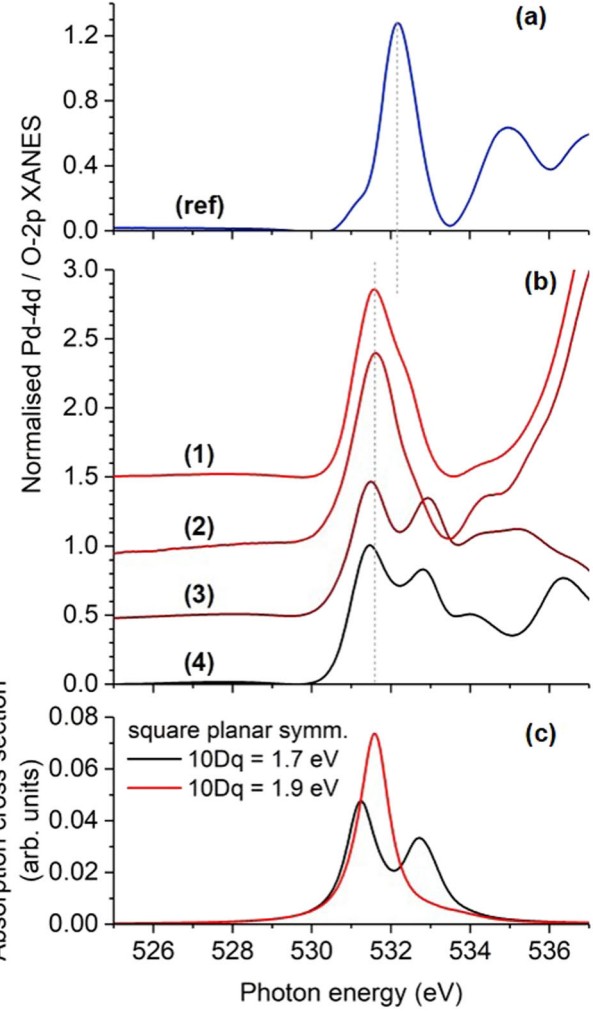

**Fig. 5 Soft X-ray absorption near edge structures of Pd.** Normalized experimental **a**, **b** and simulated **c** X-ray absorption near edge structure around the Pd $M_3$ edge of $[PdCl_4]^{2-}$ (ref), $CoPd_{12}P_8$ (1), $FePd_{12}P_8$ (2), $FePd_{12}(PhAs)_8$ (3), and $PdPd_{12}As_8$ (4). Step-like functions have been subtracted from experimental data accounting for electron excitations into higher unoccupied states than Pd $4d$ or O $2p$, respectively.

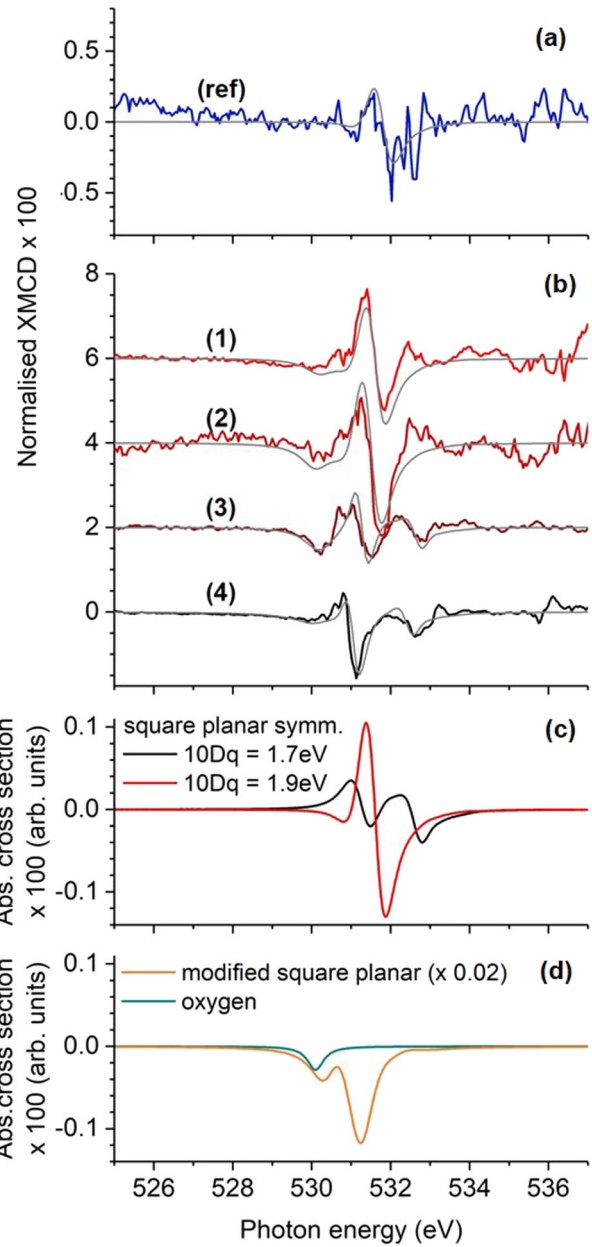

**Fig. 6 Soft X-ray magnetic circular dichroism of Pd.** Normalized experimental (**a**, **b**, thick coloured lines) and simulated **c**, **d** X-ray magnetic circular dichroism spectra around the Pd $M_3$ edge of $[PdCl_4]^{2-}$ (ref), $CoPd_{12}P_8$ (1), $FePd_{12}P_8$ (2), $FePd_{12}(PhAs)_8$ (3), and $PdPd_{12}As_8$ (4). For comparison, the best fit to experimental data is shown as thin grey lines in panels **a**, **b**. All spectra were multiplied by a factor of 100 with respect to the normalized XANES presented in Figs. 4 and 5.

All parameters are summarized in Supplementary Table 1. We start with the result for $CoPd_{12}P_8$, $FePd_{12}P_8$, and $FePd_{12}(PhAs)_8$, where all 12 palladium(II) ions are known to have the same coordination symmetry[2,3]. Here, the square-planar symmetry was introduced as a limit case of tetragonally distorted octahedral symmetry with two axial ligands at infinity. This is characterized by the relation $Dt = 2 \times 10Dq/35$[7], leaving 10Dq and Ds the only independent parameters. This rule can be derived in the ionic limit of the point charge model or by using different coupling strength ratios for the different orbitals reflected by certain electron hopping rates. The latter gives an additional relation $Ds = 19 \times 10Dq/126$[8]. With these constraints, the transition from single-peak (A) to double-peak (A1, A2) XANES can be modelled, e.g. with a slightly reduced crystal field splitting as mentioned before. The spectral XMCD differences and changes in the XMCD amplitude can also be related to this modification as presented in Fig. 6. The experimental data obtained at the Pd $M_3$ absorption edge are shown in Fig. 6a, b. In Fig. 6c, the influence of different crystal fields on the spectral shape and intensity of the simulated XMCD is presented. The negative feature a in the experimental XMCD signal was fitted by an additional Lorentzian

line (blue line in Fig. 6d) as expected from an oxygen contribution. Accordingly, the same contribution was added to the simulated XMCD spectra of $FePd_{12}P_8$ and $FePd_{12}(PhAs)_8$ and a smaller Lorentzian contribution was added to the XMCD spectrum of $CoPd_{12}P_8$.

The experimental XANES and XMCD data of $PdPd_{12}As_8$ were fitted by the same contribution of the 12 square-planar coordinated palladium(II) ions and an additional contribution of palladium(II) ions in a slightly modified crystal field, i.e. with reduced values of Dt and Ds (orange line in Fig. 6d) and a slightly increased $d$–$d$ Coulomb repulsion. The oxygen contribution to the XMCD is negligible in this case.

A comparison of simulated and experimental spectra in the whole measured energy range as well as all different contributions included are separately shown in Supplementary Fig. 3. In addition, different coordination symmetries have been tested for the central palladium(II) ion (Supplementary Fig. 4 and Supplementary Note 2). The simulations give also the possibility to further analyse the initial electronic states and different orbital contributions to the XANES fine structure and XMCD, respectively. This was partially done with the help of the CTM4DOC program that includes a graphical user interface which facilitates the representation of useful parameters returned from multiplet calculations[9]. Two cases could be clearly identified: (i) The single-peak XANES at the Pd $M_3$ absorption edge of samples (1) and (2) is related to electron transitions to the $b_1$ orbital according to an $|e^4 b_2^2 a_1^2 b_1^0\rangle$ initial state as expected for the square-planar coordination with rather large crystal field splittings and (ii) the modified crystal field around the central $Pd^{2+}$ ion of sample (4) yields an $|e^4 b_2^2 a_1^1 b_1^1\rangle$ initial electronic state, which is paramagnetic and gives rise to a two-peak fine structure in the XANES at the Pd $M_3$ absorption edge corresponding to transitions to the $b_1$ and $a_1$ orbitals. In the shell of samples (3) and (4) the two-peak XANES at the Pd $M_3$ absorption edge looks similar to the paramagnetic state, but the tiny XMCD suggests diamagnetism. Considering also the simulated linear dichroism (Supplementary Fig. 5 and Supplementary Note 3) leads to the conclusion that each of the formal $a_1$ and $b_1$ states is occupied with one electron as in the paramagnetic state, but here with opposite spins. In fact, a transition to the paramagnetic state can be induced in the simulations by increasing the molecular field, which renders a parallel alignment of the spins more favourable.

This intermediate state occurs only in systems with large spin–orbit coupling of the valence states like the $4d$ states of Pd (SOC(Pd-$4d$) ≈ 0.188 eV) which is responsible for a significant mixing of states. It is facilitated by a non-vanishing, but not too large $d$–$d$ Coulomb repulsion and intermediate crystal field splitting. In Fig. 7, these findings are illustrated. Without $4d$ spin–orbit coupling (Fig. 7a), a paramagnetic–diamagnetic transition in square-planar symmetry takes place around a normalized crystal field $10Dq/B ≈ 20$, where $B$ denotes the common Racah interelectronic repulsion parameter that can be calculated from the Slater–Condon parameters $F_{dd}^k$ according to $B = F_{dd}^2/49 - 5F_{dd}^4/441$. The atomic values $F_{dd}^2 = 7.278$ eV and $F_{dd}^4 = 4.757$ eV yield $B ≈ 94.6$ meV. For this value, the conventional diamagnetism–paramagnetism transition is at a crystal field value of about $10Dq ≈ 2.265$ eV. Additional molecular fields up to tens of meV do not show any remarkable influence of the transition position.

If the $4d$ spin–orbit coupling is included, a second diamagnetic state appears in the former paramagnetic region (Fig. 7b) that can be destabilized by molecular fields as mentioned before. In Fig. 7c the magnetic states are depicted as a function of relative changes of Ds and Dt for a fixed value of $10Dq = 1.7$ eV. For a square planar symmetry and $10Dq = 1.7$ eV, we have Dt ≈ 0.097 eV and Ds ≈ 0.256 eV. Both Dt and Ds were reduced in steps of 10% from the initial values as a measure of the degree of square planarity (or in other words, as a measure of tetragonal distortion) with limit values of 1.0 and 0 corresponding to square-planar symmetry and octahedral symmetry, respectively. In our example, a reduction of the degree of square planarity considers an increasing influence of the out-of-plane oxygen anions around the central $Pd^{2+}$ ions in $PdPd_{12}As_8$. The corresponding change in energy levels is illustrated in the panel below (Fig. 7d–f).

A deviation from square-planar symmetry destabilizes the unconventional diamagnetic state, i.e. the transition to the paramagnetic state takes already place for smaller molecular

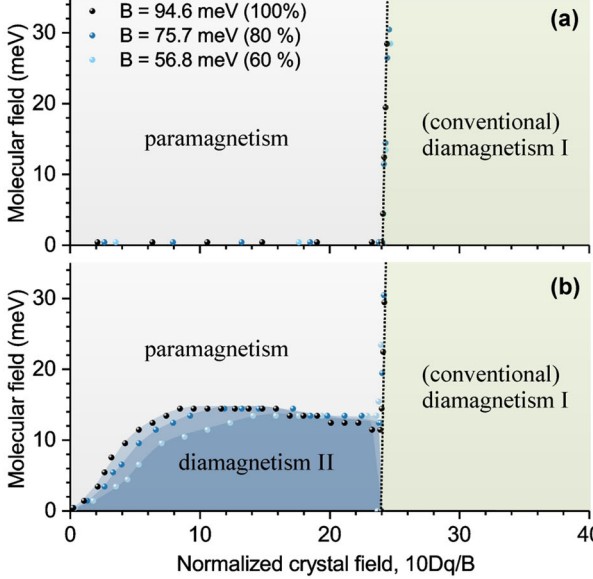

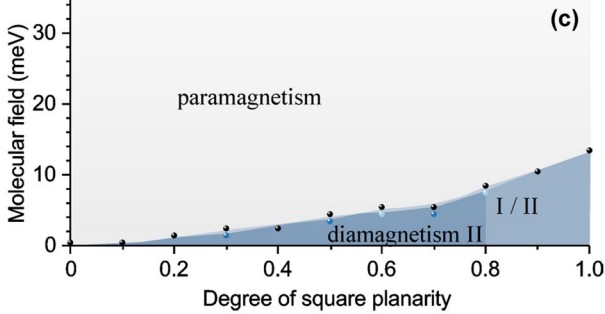

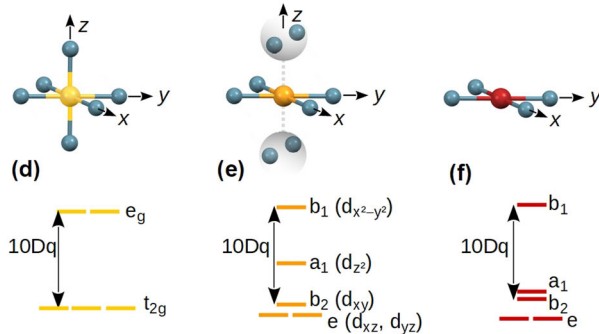

**Fig. 7 Magnetic states of palladium(II) ions.** Magnetic states of palladium (II) ions in square-planar coordination symmetry as a function of crystal field normalized to the Racah parameter $B$ without $4d$ spin–orbit coupling **a** and with $4d$ spin–orbit coupling **b**. Panel **c** shows the magnetic states as a function of degree of square-planarity by variation of Dt and Ds for a fixed value $10Dq = 1.7$ eV, where 1.0 corresponds to square-planar symmetry and 0.0 corresponds to octahedral symmetry. The molecular field is considered as a measure of stability against external parameters like, e.g. a magnetic field. Panels **d–f** illustrate the change of coordination and energy level splitting with increasing degree of square-planarity from left to right.

fields. Note that for a reduction of $B$ to 60% of the atomic value, conventional diamagnetism was obtained for high degree of square planarity above 0.8.

**Quantification of magnetic moments from experimental data.** In principle, effective spin and orbital magnetic moments can be

derived from XANES and XMCD data by a sum-rule-based analysis[10,11]. In the present case, a straight-forward quantification is not possible since the palladium absorption edges are super-imposed by the oxygen absorption edge. Thus, we follow two different approaches, i.e. (i) the determination of magnetic moments from a sum-rules-based analysis of simulated XANES and XMCD spectra fitted to experimental data and (ii) a sum-rule-based analysis of the high-quality experimental data (FePd$_{12}$(PhAs)$_8$ and PdPd$_{12}$As$_8$) with only the white-line intensity and the oxygen XMCD contribution estimated from fitted simulations.

Following the notation of Chen et al. [12], effective spin and orbital magnetic moments per absorbing atom or ion are connected to the parameters $p$ and $q$ denoting the integral of XMCD over the M$_3$ absorption edge and over both, M$_3$ and M$_2$, absorption edges, respectively:

$$m_S^{eff} = -\frac{(3p - 2q)}{r} n_h \mu_B \quad (1)$$

$$m_l = -\frac{2q}{3r} n_h \mu_B \quad (2)$$

where $r$ denotes the white-line intensity of the 4$d$ XANES, i.e. the XANES related to transitions into the 4$d$ final states that are separated from transitions into higher unoccupied states or the continuum by subtracting a two step-like function. The intensity $r$ is assumed to be proportional to the number of unoccupied final states $n_h^d$. For all palladium(II) ions we used the values of $n_h^d$ obtained from DFT as summarized in Table 2. The ratio of orbital to effective spin magnetic moment is independent of $r$ and $n_h^d$ and can be derived from the ratio $p/q$:

$$m_l/m_s^{eff} = 2/(9p/q - 6) \quad (3)$$

The integrated spectra necessary for the analysis are presented in Supplementary Figs. 6 and 7. The resulting effective spin and orbital magnetic moments can be found in Tables 2 and 3. All effective spin magnetic moments of palladium(II) ions in the shell

are in the range of $10^{-3}$–$10^{-2}\mu_B$ per atom. The orbital magnetic moments are even about one order of magnitude smaller. The central palladium(II) ion exhibits an effective spin magnetic moment of about 0.13$\mu_B$ and an orbital magnetic moment of about 0.03$\mu_B$. These large values compared to the magnetic moments of diamagnetic palladium(II) ions in the shell, clearly indicate a (Langevin) paramagnetic state.

## Discussion

The discussion is divided into three parts: In the first part, we briefly discuss the oxygen contribution to the experimental XMCD spectra, before turning to the diamagnetism of palladium (II) ions in square-planar symmetry, i.e. the palladium(II) ions in the oxopalladate shell. In the last part, we discuss the magnetism of the central palladium(II) ions as an example for the remarkable sensitivity of magnetic properties to structural modifications. For a discussion of the validity of XMCD sum rules, the interested reader is referred to the Supplementary Note 4.

The oxygen contribution to the experimental XMCD is mainly caused by hybridization effects with the central 3$d$ or 4$d$ ion that carries a sizeable magnetic moment leading to a spin (and orbital) polarization of the surrounding oxygen ions. Fitting by a single peak function is reasonable, since the XMCD of oxygen is known to exhibit a rather simple shape as reported, e.g. for the oxygen surfactant on Fe, Co, or Ni layers[13]. Another XMCD structure that can be found in the literature[14] (and in Supplementary Fig. 2), showing a negative peak followed by a positive peak, is typical for oxygen in ferrimagnetic or antiferromagnetic oxides. Since our system is paramagnetic, we do not expect such a double-peak feature.

According to the sum rule[15], the negative sign of the oxygen XMCD corresponds to a parallel alignment of the orbital magnetic moment (and the spin magnetic moment consequently) with the magnetization of the central Co$^{2+}$ or Fe$^{3+}$ ion, which is expected for these systems and in agreement to the spin-density plot (Fig. 3a) shown for the case of CoPd$_{12}$P$_8$. Although the hybridization with 4$d$ elements like palladium is usually stronger because of the more delocalized character of the 4$d$ electrons

**Table 2 Magnetic moments of Pd as estimated from simulated spectra.**

| | Sample | Pd position | $n_h^d$ | $m_s^{eff}$(Pd) ($10^{-2}$ $\mu_B$/atom) | $m_l$(Pd) ($10^{-2}$ $\mu_B$/atom) | $m_l/m_s^{eff}$ (%) | $m_{tot}$ (Pd) ($10^{-2}$ $\mu_B$/atom) |
|---|---|---|---|---|---|---|---|
| (ref) | [PdCl$_4$]$^{2-}$ | | 2 | ≤0.25 | ≤0.05 | 27 ± 5 | ≤0.3 |
| (1) | CoPd$_{12}$P$_8$ | Shell | 1.46 | 0.8 ± 0.2 | 0.23 ± 0.06 | 27 ± 8 | 1.0 ± 0.2 |
| (2) | FePd$_{12}$P$_8$ | Shell | 1.47 | 1.0 ± 0.2 | 0.28 ± 0.06 | 27 ± 8 | 1.2 ± 0.2 |
| (3) | FePd$_{12}$(PhAs)$_8$ | Shell | 1.47 | 0.15 ± 0.08 | −0.04 ± 0.02 | −24 ± 10 | 0.11 ± 0.03 |
| (4) | PdPd$_{12}$As$_8$ | Shell | 1.46 | 0.15 ± 0.08 | −0.04 ± 0.02 | −24 ± 10 | 0.11 ± 0.03 |
| | | Centre | 1.70 | 13 ± 10 | 2.7 ± 2 | 21 ± 8 | 15 ± 10 |
| | | Averaged | | 1.1 ± 0.5 | 0.18 ± 0.15 | 11 ± 9 | 1.3 ± 0.5 |

Estimated relative effective spin and orbital contributions to the magnetic moment assigned to palladium as well as the ratio of orbital-to-spin magnetic moment and total magnetic moments per palladium(II) ion in 6 T at 4.3–4.6 K. From simulated spectra, magnetic moments in Bohr magnetons have been quantified for Pd ions in the shell and central Pd ions, respectively, using estimated numbers of unoccupied $d$ states $n_h^d$ from DFT. For the reference sample [PdCl$_4$]$^{2-}$, the free-ion value $n_h^d = 2$ has been used.

**Table 3 Magnetic moments of Pd as determined by sum-rules based analyses of experimental data.**

| | Sample | Pd position | $n_h^d$ | $m_s^{eff}$(Pd) ($10^{-2}$ $\mu_B$/atom) | $m_l$ (pd) ($10^{-2}$ $\mu_B$/atom) | $m_l/m_s^{eff}$ (%) | $m_{tot}$ (pd) ($10^{-2}$ $\mu_B$/atom) |
|---|---|---|---|---|---|---|---|
| (3) | FePd$_{12}$(PhAs)$_8$ | Shell | 1.47 | 0.06 ± 0.04 | −0.04 ± 0.01 | −63 ± 15 | 0.02 ± 0.02 |
| (4) | PdPd$_{12}$As$_8$ | Averaged | 1.48 | 1.0 ± 0.2 | 0.33 ± 0.10 | 35 ± 9 | 1.3 ± 0.2 |

Estimated relative effective spin and orbital contributions to the magnetic moment assigned to palladium as well as the ratio of orbital-to-spin magnetic moment and total magnetic moments per palladium(II) ion in 6 T at 4.3–4.6 K as extracted from experimental spectra using estimated numbers of unoccupied $d$ states $n_h^d$ from DFT. The white line intensity was estimated from simulated data.

compared to $3d$ electrons, no clear indication of an oxygen XMCD was obtained in $PdPd_{12}As_8$. This can be explained by the smaller spin magnetic moment of $Pd^{2+}$, i.e. about $0.13\mu_B$ as estimated in this work, while the $Co^{2+}$ ($Fe^{3+}$) carries a magnetic saturation moment of about $3\mu_B$ ($5\mu_B$)[4]. Together with the size-able XMCD signal from the Pd at approximately the same photon energy, an oxygen XMCD cannot be resolved in the paramagnetic state of $PdPd_{12}As_8$.

In square-planar oxygen coordination, palladium(II) ions are known to exhibit low-spin states. Due to their $d^8$ electron configuration, the low-spin state is characterized by $S = 0$ and, consequently, the system shows a diamagnetic response to magnetic fields. The scheme of energy levels (in the one electron approximation) is presented in Fig. 7f. If the energy difference between the highest ($b_1$) and second highest ($a_1$) level is larger than the spin pairing energy, the low spin $d^8$ electron configuration is preferred in this simplified picture. Like the Coulomb repulsion, the spin pairing energy is usually smaller for $4d$ elements compared to $3d$ elements, because the $4d$ orbitals are more delocalized. This connection is also the reason, why the conventional diamagnetic–paramagnetic transition can also be reached in multiplet calculations by changing the Coulomb repulsion (Slater–Condon or corresponding Racah parameters) for the $4d$ electrons. As shown in Fig. 7a, b the transition depends on the ratio of crystal field splitting and Racah interelectronic repulsion parameter, 10 Dq/B. Moreover, it can be triggered by ligand-to-metal charge transfer or temperature effects. Thus, there exist other sets of possible parameters describing both the different spectral shapes of XANES and XMCD in square-planar coordination as well as the occurrence of the Langevin paramagnetic state. Therefore, the parameters used for the simulations as summarized in Supplementary Table 1 are only one reasonable example for describing the experimental results while keeping the number of fitting parameters as small as possible. However, in this work we use the simulations for a qualitative description of the XANES and XMCD fine structures and to support extraction of magnetic moments from experimental data. A further analysis of charge transfer and delocalization effects is beyond the scope of this paper.

For the discussion of magnetic moments by sum-rules based analysis, the influence of the number of unoccupied $d$ states, $n_h^d$, has to be considered. In general, values of $n_h^d$ computed by DFT methods should be preferred to the electron counts of a free ion, since they include hybridization effects. In this work $n_h^d$ was obtained from reduced Mulliken charges and depends on the basis set. Although the values are error-prone, a qualitative comparison is possible and the quantitative numbers for the rather localized $d$ states (with respect to $s$ states) are considered to be a more reasonable approximation than free-ion values.

For $CoPd_{12}P_8$ and $FePd_{12}P_8$ the palladium(II) ions in the shell have tiny total magnetic moments in the order of $10^{-2} \mu_B$ with a large ratio of orbital to effective spin magnetic moment of 27% as determined from a sum-rules-based analysis of simulated spectra fitted to the experimental data. Moreover, from the XMCD amplitudes it is obvious that the total magnetic moments are larger than the one of the $[PdCl_4]^{2-}$ reference sample. This may indicate less localized states in the palladate samples since a diamagnetic response scales with the electron "orbit" $\langle r^2 \rangle$ in the classical Langevin model (which gives essentially the same predictions as quantum theory).

The XMCD fine structure that was obtained for the diamagnetic palladium(II) ions in the shell of $FePd_{12}(PhAs)_8$ reminds of the XMCD signal of Au measured in the hard X-ray regime at the $L_{3,2}$ absorption edges (also $p$ to $d$ transitions) reported by Suzuki et al.[16]. In their work, the authors focus on Pauli and orbital paramagnetism of the $5d$ electrons in line with a prominent $5d$ contribution to the conduction electrons[17] and connected to a large orbital magnetic moment induced via spin–orbit coupling. In our case, a possible paramagnetic contribution at the $M_3$ absorption edge would be superimposed by the oxygen XMCD. A sum-rule-based analysis of both simulated and experimental spectra yields an orbital magnetic moment aligned antiparallel to the external magnetic field. The ratio of orbital to effective spin magnetic moment is $-25\%$ as determined from simulated spectra and $-63\%$ from experimental spectra. The large absolute value also agrees to the interpretation of Suzuki et al. that for the case of XMCD measured of gold metal "the structures originate from the very strong orbital character of the higher-order [$s$ and $d$] empty states"[16]. In our case it turned out that the mixing of states due to the large $4d$ spin–orbit coupling is responsible for this modified diamagnetic state. In this regard, the intermediate diamagnetic state obtained is another example of the importance of spin–orbit coupling discussed e.g. for the case of iridates[18].

Since the oxopalladate shells in the four samples are very similar as mentioned above and their tiny XMCD signal is related to a diamagnetic initial state, the paramagnetic contribution that was found in conventional magnetometry of $PdPd_{12}As_8$ (Fig. 2) is expected to originate from the palladium(II) ion in the centre. This is in agreement with the spin-density plot (Fig. 3) obtained from DFT calculations. Formally, this ion is also located in a square-planar coordinated oxygen environment[1]. However, the additional oxygen ions in close vicinity modify already the crystal field of the square-planar coordinated palladium(II) ion so that Ds and Dt are slightly reduced. This leads to a change in the crystal field splitting as sketched in Fig. 7e, i.e. the energy difference between $a_1$ and $b_1$ levels is reduced, enabling and explaining the paramagnetism observed in our experiments.

As presented in Supplementary Fig. 4 and described in Supplementary Note 2, the different contributions to the XMCD at $M_3$ and $M_2$ edges assuming higher symmetries and/or coordination numbers, i.e. octahedral symmetry with six-fold coordination or eight-fold coordination in cubic or tetrahedral symmetry, are remarkably different and do not provide good or reliable fitting results, respectively, as quantified by the corresponding sum of squared residuals summarized in Supplementary Table 2. Again, this finding is in line with DFT calculations that show a six-fold coordination of the central palladium ion (and two additional oxygen anions at a large distance of 2.7 Å), but not an octahedral symmetry, after geometric optimization starting from a square-planar symmetry of the central ion.

Interestingly, it can be deduced from multiplet calculations that the paramagnetic response in this region close to square-planar symmetry can be induced, e.g. by external magnetic fields (Fig. 7) since it leads to a further separation of magnetic substates favouring a parallel alignment of spins. This example shows that the application of large external magnetic fields in order to measure a magnetic response, may already alter the properties of the system. Furthermore, paramagnetism has already been reported for higher symmetries and/or coordination numbers of the central palladium(II) ion achieved by replacing the $As^VO_3$ fragments outside the oxopalladate shell by either $Se^{IV}O_3$ fragments or $PhAs^VO_3$ resulting in six-fold octahedral or eight-fold coordination, respectively[19].

In conclusion, our results show that already small modifications of the local environment may change the magnetic state of palladium(II) ions from diamagnetic to dominating paramagnetic marking an intramolecular crossover when the square-planar coordination symmetry is extended by additional out-of-plane anions. This finding demonstrates the importance of the local environment—not only nearest-neighbour atoms or ions—in the structural input for calculating physical properties. In the

transition region, the large $4d$ spin–orbit coupling gives rise to an intermediate diamagnetic state, in which antiparallel oriented spins occupy formally different orbitals ($b_1$ and $a_1$). This state can be destabilized e.g. by magnetic fields leading again to the paramagnetic state.

Since the soft X-ray regime is nowadays also accessible, e.g. by higher harmonic generation (HHG)[20], the different XANES fine structures of the two diamagnetic states can also be measured using lab-sources and may stimulate further investigations of dynamic electronic characteristics of the unconventional diamagnetic state.

The detection of the small XMCD signal in diamagnetic states became possible with the high brilliance of third generation synchrotron sources combined with an end station offering high magnetic fields and low sample temperatures and is the first example of XMCD from a diamagnet in the soft X-ray regime. In this way the XANES/XMCD method gives important insight to the electronic structure and may become a useful characterization tool for other diamagnetic materials like e.g. the surface region of superconductors or certain topological insulators.

## Methods

**Notation**. Throughout this paper, crystal fields are described by the parameters 10Dq, Dt, and Ds. Both Dt and Ds yield a further splitting of the degenerated energy levels of an octahedral crystal symmetry from $t_{2g}$ ($d_{xz}$, $d_{yz}$, $d_{xy}$) and $e_g$ ($d_{z2}$, $d_{x2-y2}$) to $e$ ($d_{xz}$, $d_{yz}$), $b_2$ ($d_{xy}$), $a_1$ ($d_{z2}$), and $b_1$ ($d_{x2-y2}$). Note, this scheme of energy level splitting is strictly valid only for one electron ($d^1$) systems that can be described by a single electron wave function and is merely used here for illustration.

**Sample preparation**. The polyoxopalladates (1)–(3) were synthesized in a one-pot reaction of [Pd$_3$(CH$_3$OO)$_6$] and an appropriate salt of the $3d$ metal dissolved either in sodium phosphate buffer to obtain the phosphate-capped polyoxopalladate samples (1) and (2) or with phenylarsonic acid in sodium acetate buffer solution to obtain the phenyl arsonate-capped polyoxopalladate sample (3). For sample (4) with palladium(II) ions at the central position and in the shell, PdCl$_2$ was used as precursor dissolved with As$_2$O$_5$ in sodium acetate buffer solution. More details like temperatures and adjustment of pH values are described elsewhere[1–3]. The solution stabilities were demonstrated by (multi-)nuclear magnetic resonance spectroscopy[2,3]. After crystallization, their identities were confirmed by infra-red and single-crystal X-ray diffraction measurements, For X-ray absorption measurements, the dried polyoxopalladate powder samples were dissolved in water with a concentration in the range of $10^{-4}$ mol/l and drop-coated onto a freshly cleaved HOPG substrate. Heating to 50 °C accelerated the evaporation of the water without thermal decomposition of the molecules. After the drying process, the electric contact between sample surface and sample holder was improved by silver paste added on the sample edges. The in-plane electric conductivity of HOPG is very high and no visible charging effects occurred when measuring in the sample centre. Neither background instabilities, nor discharging spikes, nor artificial line broadening related to TEY artifacts were observed. Photographs of the samples are shown in Supplementary Fig. 1.

**Magnetometry**. Conventional magnetometry measurements were carried out on powder samples using a Magnetic Properties Measurement System (Quantum Design) in magnetic fields up to 5 T and temperatures down to 2 K. The magnetic moment per cluster was calculated from the total magnetic moment by using the mass of the sample measured by a microbalance and the well-known molecular masses.

**X-ray absorption spectroscopy**. X-ray absorption spectra were measured in the high-field end station of the beamline UE46-PGM1 of the Helmholtz-Zentrum Berlin BESSY II synchrotron source at $T = 4.3$–$4.5$ K in magnetic fields of 6 T. The absorption was detected in total electron yield (TEY) mode by measuring the sample drain current. The XMCD was obtained by measuring several sets of four spectra with different helicity (positive, negative, negative, positive) in a constant external magnetic field. Afterwards, the external magnetic field was reversed and the measurements were repeated.

**Charge transfer multiplet calculations**. Charge transfer multiplet calculations were done using the CTM4XAS program package[5]. A molecular field of 10 meV was introduced to account for the magnetization. More details and a summary of all parameters used for the simulations presented here can be found in Supplementary Table 1.

**DFT calculations**. DFT calculations were performed using the B3LYP hybrid functional with def2-TVZP[21], def2/J[22] basis sets as implemented in the ORCA package[23]. To reduce computation time, the auxiliary RIJCOSX basis set was employed[24]. For the case of palladium ions, an effective core potential (ECP)[25] was used which is known to give reasonable results for structural optimization and energy calculations as presented here. As convergence limits, default values were used, i.e. for the energy change $5 \times 10^{-6}E_h$, where $E_h$ denotes the Hartree energy, the maximum force was limited to $3 \times 10^{-4}E_h/r_B$, average (root mean square) force on all atoms: $1 \times 10^{-4}E_h/r_B$, maximum displacement in the last two iterations: $4 \times 10^{-3}r_B$, average (root mean square) displacement in the last two iterations: $2 \times 10^{-3}r_B$.

## Data availability
The authors declare that all data supporting the findings of this study are available within the paper and its supplementary information files. The experimental source data for Figs. 2–5 are available from the corresponding author upon reasonable request.

## Code availability
The CTM4XAS program package used for atomic multiplet calculations is freely available after registration. http://www.anorg.chem.uu.nl/CTM4XAS/software.html. The binaries of ORCA are available free of charge for academic users after registration. https://orcaforum.kofo.mpg.de/app.php/portal.

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

## Acknowledgements

We thank the Helmholtz-Zentrum Berlin (HZB) for the allocation of synchrotron radiation beamtime and access to the laboratory for magnetic measurements of the CoreLab Quantum Materials. For kind support we thank the HZB staff, particularly E. Weschke and E. Schierle. J. van Leusen (RWTH Aachen University) is gratefully acknowledged for additional magnetometry measurements and helpful discussions. This work was partly funded by the Helmholtz Association (Young Investigator's Group Borderline Magnetism under contract no. VH-NG-1031).

## Author contributions

C.S.-A., A.S., D.S., and S.F.S. performed the X-ray absorption experiments, K.S. and C.S.-A. the conventional magnetometry measurements. C.S.-A. and A.S. analysed the data, C.S.-A. and F.M.f.d.G. performed and interpreted charge transfer multiplet simulations, C.S.-A. performed DFT calculations. N.V.I. and M.S. synthesized the polyoxopalladates. S.F.S. synthesized the ferrite reference sample. C.S.-A., P.K., A.S., and D.S. wrote the paper. All authors contributed to the scientific discussion of the results and commented on the manuscript.

## Competing interests

The authors declare no competing interests.
