## [Peer Review File · Communications Chemistry]

Reviewers' comments:

Reviewer #1 (Remarks to the Author):

This manuscript discusses the diamagnetic to paramagnetic spin crossover in Pd ions. The conclusion are supported by magnetic measurements and XMCD. The title claims that the transition is spin-orbit coupling driven, but the case for that appears to be rather weak. The dominant driving force appears to be "small modifications of the local environment [that] change the magnetic state of palladium(II) ions from dia- to dominating paramagnetism." Now while it is true that "In the transition region, the large 4d spin-orbit coupling gives rise to an intermediate diamagnetic state, in which antiparallel oriented spins occupy formally different orbitals," this does not appear to make it the driving force, in my opinion. Therefore, we are left with a spin transition as a result of a competition between Hund's rule Coulomb interactions and crystal field. However, this is similar to the majority of the spin transitions. So I am not convinced that this is special enough to warrant publication in Nat. Chem. Additionally, there appear to be few broader impacts of the findings. That being said, the paper is well written and an important piece of research.

Reviewer #3 (Remarks to the Author):

Authors have reported experimental XMCD results about palladium(II) ions in polyoxopalladates, PdPd₁₂As₈, FePd₁₂P₈, FePd₁₂(PhAs)₈, and CoPd₁₂P₈. They used conventional magnetometry and simulations based on charge transfer multiplet calculations. The manuscript is well written and presented. The main text and supplementary information files are very rich and constructive. I have few questions regarding the following points :

1. Why authors didn't report multinuclear NMR spectroscopy to investigate the solution stability of the diamagnetic polyanions PdPd₁₂As₈, FePd₁₂P₈, FePd₁₂(PhAs)₈, and CoPd₁₂P₈.
2. Does palladium(II) ions in polyoxopalladates have other derivatives than PdPd₁₂As₈, FePd₁₂P₈, FePd₁₂(PhAs)₈, and CoPd₁₂P₈ ? are these samples stable mechanically and energetically ?
3. Why authors have opted to use the CTM4XAS simulations to compute their different spectra than density functional theory (considering the spin-orbit coupling). Furthermore, the crystal field splitting as well as energy differences between the involved orbitals can also be calculated from DFT.
4. Authors are claiming that electronic structure may become a useful characterization tool for other diamagnetic materials like e.g. the surface region of superconductors or certain topological insulators. It would be important to complete the theoretical work with the electronic band structure and spin-densities maps to confirm the diamagnetism/ paramagnetsim changes of the present compounds: PdPd₁₂As₈, FePd₁₂P₈, FePd₁₂(PhAs)₈, and CoPd₁₂P₈.
5. How the magnetic moments (total and by element) compare in different samples (table 2) and also to similar materials reported in literature?
6. We would like to see more discussion of the results in light of transition between 2D to 3D and how it affect changes of electronic and magnetic properties (diamagnetism vs paramagnetism).
7. How the temperature can affect the present observations and conclusions?

Minor correction

1. English has to be polished such as: "electronic structure and ? may become a useful characterization tool for other diamagnetic materials like e.g. the surface region of superconductors or certain topological insulators". What is ? (I guess magnetic)
2. The abstract needs to be reformulated and made more clear and attractive for the reader. We don't understand the purpose of the present research when we read it.
3. Some figures need to be plotted clearly, such as those reported in supplementary information.

Spin-orbit coupling driven intramolecular crossover from 2D diamagnetism to 3D paramagnetism of Pd ions

Alevtina Smekhova et al.

This manuscript reports the detailed analysis of Pd M-edge XMCD for Pd complexes comparing with the simulations. In spite of the efforts of authors, I cannot recommend to the publication from high impact journal. Mainly, the title and conclusions are not supported from the results. First, 'spin-orbit driven' in the title is only the use of Pd ions at the center, which is obvious. Second, I cannot find crossover phenomena between dia- and para-magnetism. Third, overlapping of Pd M-edge and O K-edge from surface contaminations is serious problem. For the discussion of Pd compounds including oxygen, Pd L-edge should be used and straightforward. Otherwise, I cannot find whether the present data is artifact or not without surface sensitive total-electron yield mode. Therefore, unfortunately, this manuscript does not provide important messages for the readers of high impact journal 'Communications Chemistry'. On the other hand, the analysis of Pd with simulation is excellent and interesting. This effort will be opened in cases of oxygen-free defined Pd compounds.

Followings are the questions and comments by reading the manuscript.

1. Magnetic moments from the magnetometry are not explained by the XMCD. All curves in Fig.2 are paramagnetism. I cannot understand the definition of para- and dia-magnetism in this manuscript.
2. XMCD of Co and Fe should be presented for (1) and (2) compounds corresponding to above comment 1.
3. The challenges for deconvolution of Pd signals are excellent. However, the process how Pd contribution is estimated is still unclear. Considering the ionization cross-section of Pd M-edge, the estimations might become different.
4. For the sum rule analysis, simulated spectra are used. Deconvolution of two sites in Fig. S2(3)-(4) is excellent. However, the authors use single peak for XAS in (1) but there should be satellite structures of 3p->5s transition. Check other Pd M-edge line shapes (Sci. Rep. 8. 8303, 2018). Therefore, simulations do not detect the true Pd spectral line shapes.
5. Error bars are required.
6. In order to avoid ambiguities for estimations, the detections by Pd L-edge are required. Otherwise, the analysis in this manuscript is not guaranteed.

We thank the reviewers for their helpful comments and criticism. In particular suggesting DFT calculations turned out very useful and definitely helped us to improve the manuscript and substantiate our conclusions. Besides adding DFT results, we also made changes to the title and abstract to emphasise the main message of our work. A list of major changes and detailed point-by-point answers are given below, relevant changes in the manuscript are highlighted in red. Minor changes like typo corrections and transitional phrases are not explicitly marked for clarity.

List of major changes

1. The paper's title was rephrased to more precisely convey the central aspects of this work.
2. The abstract was rewritten and shortened to ~150 words.
3. DFT (B3LYP) calculations were included (page 5, 6; new Figure 3; new references 21-25).
4. Figure 6 was modified and included in the new Figure 7 to meet the limit of max. 10 displayed items.
5. A new Figure 7 was added showing the different magnetic states and the influence of spin-orbit coupling and crystal field modifications to illustrate the results mentioned in the text (Figure 7 and additional text on page 10).
6. As an outlook, the advantage of measurements in the soft x-ray regime (M edges) was added, in perspective to the increasing number of lab sources for dynamic studies in this regime (page 16, new reference 20).
7. Data availability and code availability statements were added.

Answer to Reviewer #1:

"This manuscript discusses the diamagnetic to paramagnetic spin crossover in Pd ions. The conclusion are supported by magnetic measurements and XMCD. The title claims that the transition is spin-orbit coupling driven, but the case for that appears to be rather weak. The dominant driving force appears to be "small modifications of the local environment [that] change the magnetic state of palladium(II) ions from dia- to dominating paramagnetism." Now while it is true that "In the transition region, the large 4d spin-orbit coupling gives rise to an intermediate diamagnetic state, in which antiparallel oriented spins occupy formally different orbitals," this does not appear to make it the driving force, in my opinion."

- ▶ We fully agree and removed the phrase "spin-orbit driven". In the revised manuscript, it is more precisely phrased "facilitated by spin-orbit coupling" and the new Figures 7a, b illustrate the influence of 4d spin-orbit coupling.

"Therefore, we are left with a spin transition as a result of a competition between Hund's rule Coulomb interactions and crystal field. However, this is similar to the majority of the spin transitions."

- ▶ We regret that in our first draft the main findings were not made sufficiently clear. Concerning the transition, we emphasise in the revised version that there are two different diamagnetic states, e.g. by illustrating the different states and the transitions in Figure 7. The influence of the additional out-of-plane anions leads to a transition from the unconventional diamagnetic state to the common paramagnetic state (Fig. 7c). We also changed the title accordingly "...unconventional diamagnetism to paramagnetism...".

"So I am not convinced that this is special enough to warrant publication in Nat. Chem. Additionally, there appear to be few broader impacts of the findings."

- ▶ We agree that it is not suitable for publication in Nature Chemistry. But it was submitted to Communications Chemistry that publishes also research of interesting to smaller, more specialized communities. In our case, we believe that the work is very interesting e.g. for the XMCD community since we present the very first example of XMCD of a diamagnet and also addresses the molecular magnetism community since we showcase an example where the magnetic state cannot be concluded from structural analysis in a straightforward manner. In the revised version, we stressed this aspect a bit more.

“That being said, the paper is well written and an important piece of research.”

Answer to Reviewer #2:

“This manuscript reports the detailed analysis of Pd M-edge XMCD for Pd complexes comparing with the simulations. In spite of the efforts of authors, I cannot recommend to the publication from high impact journal. Mainly, the title and conclusions are not supported from the results.”

- ▶ The title has been changed to better fit the results presented in the manuscript. Unfortunately, the referee does not clarify which point of the conclusions is not supported.

“First, ‘spin-orbit driven’ in the title is only the use of Pd ions at the center, which is obvious.”

- ▶ The phrase “spin-orbit driven” has been removed from the title and the whole manuscript in order to avoid confusion. In the revised manuscript, we refer to “facilitated by spin-orbit coupling”; the newly added Figures 7a, b illustrate the influence of 4d spin-orbit coupling.

“Second, I cannot find crossover phenomena between dia- and para-magnetism.”

- ▶ The different diamagnetic and paramagnetic states are illustrated in the new Figure 7a-c. We hope that in particular Figure 7c helps to clarify the crossover between unconventional diamagnetism and paramagnetism described in the text. Please note that the term “crossover phenomena” has not been mentioned in the manuscript.

“Third, overlapping of Pd M-edge and O K-edge from surface contaminations is serious problem. For the discussion of Pd compounds including oxygen, Pd L-edge should be used and straightforward. Otherwise, I cannot find whether the present data is artifact or not without surface sensitive total-electron yield mode. Therefore, unfortunately, this manuscript does not provide important messages for the readers of high impact journal ‘Communications Chemistry’. On the other hand, the analysis of Pd with simulation is excellent and interesting. This effort will be opened in cases of oxygen-free defined Pd compounds.”

- ▶ In our case, the oxygen K edge signal is related to the molecular sample and not to surface contaminations. It contains roughly three times more oxygen than palladium.
- ▶ We agree that a sum-rule based analysis of magnetic moments is easier at the Pd $L_{3,2}$ edges. But this was not the scope of the present study. In the revised manuscript we summarise our results more clearly: (i) an unusual diamagnetic state of palladium was identified mainly from the XMCD fine structure, which is certainly not an artefact and (ii) we present, for the first time, XMCD of a diamagnet in the soft x-ray regime. The determination of magnetic moments is just a minor point here and was added for the sake of completeness.
- ▶ In the case of oxygen-free compounds, investigations at the Pd $M_{3,2}$ edges are already established. In this regard, we demonstrate that careful analysis even allows for investigation of compounds containing oxygen.

"Followings are the questions and comments by reading the manuscript.

1. Magnetic moments from the magnetometry are not explained by the XMCD. All curves in Fig.2 are paramagnetism. I cannot understand the definition of para- and dia-magnetism in this manuscript."

- ▶ We use the common definitions of paramagnetism and diamagnetism. The motivation for the XMCD study was its element selectivity (and even site specificity) in contrast to magnetometry as it was mentioned on page 4 related to sample (4). For clarity, we added on page 4 after description of magnetometry data of samples (1)-(3): "Consequently, conventional magnetometry measurements are dominated by the paramagnetic response of the central Fe and Co ion".

"2. XMCD of Co and Fe should be presented for (1) and (2) compounds corresponding to above comment 1."

- ▶ XANES, XMCD and XMLD data as well as magnetic moments for these compounds have been published elsewhere (former reference 16, now reference 4). We added a corresponding statement (page 4): "It has already been evidenced experimentally [4] that the magnetism of Fe and Co central ions in polyoxopalladates can be well described within this model."

"3. The challenges for deconvolution of Pd signals are excellent. However, the process how Pd contribution is estimated is still unclear. Considering the ionization cross-section of Pd M-edge, the estimations might become different."

- ▶ The ionization cross section of the Pd M edge and the oxygen K edge are not known accurately enough to provide a detailed fit of their relative contribution, at least not as accurate as fitting the experiment as has been done in the manuscript.
- ▶ The magnetic moments of central Pd ions and Pd ions in the shell were separated and quantified in Bohr magnetons by using the simulated spectra (see orange and red lines, respectively, in Fig. S2).

"4. For the sum rule analysis, simulated spectra are used. Deconvolution of two sites in Fig. S2(3)-(4) is excellent. However, the authors use single peak for XAS in (1) but there should be satellite structures of 3p->5s transition. Check other Pd M-edge line shapes (Sci. Rep. 8. 8303, 2018). Therefore, simulations do not detect the true Pd spectral line shapes."

- ▶ In the work mentioned, the authors studied Co/Pd multilayers with metallic Pd. Also other works using XANES/XMCD at the Pd M edges deal with metallic Pd, while we investigated Pd²⁺ ions. For instance, after reduction of Pd by hydrogen treatment, we can see the satellite structure as well, but this is beyond the scope of this paper.

"5. Error bars are required."

- ▶ We could not find missing error bars except in Table 1, where we now added them (variation of bond lengths), "which are much larger than the experimental uncertainties".

"6. In order to avoid ambiguities for estimations, the detections by Pd L-edge are required. Otherwise, the analysis in this manuscript is not guaranteed."

- ▶ See answer to point "third" above. In addition, we added a sentence in the outlook part as to why in particular M edges are interesting: "Since the soft x-ray regime is nowadays also accessible e.g. by higher harmonic generation (HHG) [20], the different XANES fine structures of the two diamagnetic states can also be measured using lab-sources and may stimulate further investigations of dynamic electronic characteristics of the unconventional diamagnetic state."

Answer to Reviewer #3:

“Authors have reported experimental XMCD results about palladium(II) ions in polyoxopalladates, PdPd₁₂As₈, FePd₁₂P₈, FePd₁₂(PhAs)₈, and CoPd₁₂P₈. They used conventional magnetometry and simulations based on charge transfer multiplet calculations. The manuscript is well written and presented. The main text and supplementary information files are very rich and constructive. I have few questions regarding the following points :

1. Why authors didn't report multinuclear NMR spectroscopy to investigate the solution stability of the diamagnetic polyanions PdPd₁₂As₈, FePd₁₂P₈, FePd₁₂(PhAs)₈, and CoPd₁₂P₈.”

- ▶ The solution stability had already been previously demonstrated and reported in the supporting data of papers focusing on the chemical synthesis of the samples. For instance, ¹³C NMR and ¹H NMR for compounds with phenylarsonate capping groups and ³¹P NMR for phosphate capping groups. We added a sentence to the methods section accordingly: “(...), the solution stabilities were demonstrated by (multi-)nuclear magnetic resonance spectroscopy.^{2,3}”

“2. Does palladium(II) ions in polyoxopalladates have other derivatives than PdPd₁₂As₈, FePd₁₂P₈, FePd₁₂(PhAs)₈, and CoPd₁₂P₈ ? are these samples stable mechanically and energetically ?”

- ▶ Yes, there are other derivatives. For instance, polyoxopalladates with palladium(II) central ions and phenylarsonate or selenate capping groups as briefly mentioned in the discussion (page 16, line 6). Furthermore, polyoxopalladates with central ions of almost every transition metal or lanthanide also exist [Chem. Eur. J. 16, 9076 (2010)]. The samples are rather stable, with the polyanions starting to decompose at temperatures around 200 °C as shown by thermogravimetric analysis. Since all these investigations are presented in the corresponding synthesis papers (and their supporting materials) , we did not replicate them in our present work.

“3. Why authors have opted to use the CTM4XAS simulations to compute their different spectra than density functional theory (considering the spin-orbit coupling). Furthermore, the crystal field splitting as well as energy differences between the involved orbitals can also be calculated from DFT.”

- ▶ The reason for using a model Hamiltonian (multiplet) code such as CTM4XAS is that codes based on DFT do not take the final state 3p core hole into account, other than its potential. the two-electron integrals coupling the 3p and 4d electrons are ~5 eV in magnitude and these interactions are neglected in DFT (and also (largely) in TD-DFT). Because the two-electron integrals coupling a p core state with a d valence state are large, all DFT and TD-DFT (or BSE) calculations break down and these two-electron integrals have to be included for an accurate description of 2p and 3p core states in 3d and 4d systems (see e.g. [Coord. Chem. Rev. 249, 31 (2005)]).
- ▶ The reviewer is right in that DFT can be used to calculate crystal field splittings that can be subsequently used in multiplet codes. However, the extraction of crystal field parameters is not trivial and error-prone and since we do not aim at the determination of the exact crystal field parameters, we decided not to follow up on this suggestion.

“4. Authors are claiming that electronic structure may become a useful characterization tool for other diamagnetic materials like e.g. the surface region of superconductors or certain topological insulators. It would be important to complete the theoretical work with the electronic band structure and spin-densities maps to confirm the diamagnetism/ paramagnetsim changes of the present compounds: PdPd₁₂As₈, FePd₁₂P₈, FePd₁₂(PhAs)₈, and CoPd₁₂P₈.”

- ▶ We thank the referee for the suggestion to include calculated spin-density maps (Figure 3). The results are in agreement to our previous interpretations, i.e. (i) the paramagnetism of PdPd₁₂As₈ is related to the central Pd ion and not to the Pd ions in the

surrounding cluster shell and (ii) the coordination of the central Pd ion can be described by a six-fold coordinated low symmetry. In the new Figure 3, we show for comparison the spin-density plot for $\text{CoPd}_{12}\text{P}_8$. $\text{FePd}_{12}\text{P}_8$ has also been calculated (but is not shown here) and exhibits the expected almost spherical spin density of the $3d^5$ high-spin configuration. Full details of our DFT studies will be presented in a separate paper.

“5. How the magnetic moments (total and by element) compare in different samples (table 2) and also to similar materials reported in literature?”

- ▶ We could not find magnetic moments reported on similar materials. In the literature on palladium(II) complexes, the majority of samples is diamagnetic and no magnetisation data are presented or analysed. For the case of paramagnetic palladium(II) complexes, we found only saturation magnetic moments reported (around $2\mu_B$), which is unfortunately insufficient for our discussion. For other systems, like metallic Pd with 3d impurities, the magnetic moments of our paramagnetic Pd is in the same order of magnitude, but a factor of 3-4 smaller, which seems to be reasonable.

“6. We would like to see more discussion of the results in light of transition between 2D to 3D and how it affect changes of electronic and magnetic properties (diamagnetism vs paramagnetism).”

- ▶ In the new Figure 7c we illustrate the paramagnetic-diamagnetic transition as a function of degree of square planarity, i.e. for pure square-planar coordination symmetry (1.0) and increasing influence of additional out-of-plane anions until the system reaches octahedral symmetry (0.0). A discussion is added on page 10.
- ▶ From a language point of view, the dimensionality has retreated into the background, since it may lead to confusion from a formal mathematical or theoretical physics point of view. Therefore, 2D and 3D was also removed from the title.

“7. How the temperature can affect the present observations and conclusions?”

- ▶ In the previous draft, in fact we mentioned that the diamagnetic-paramagnetic transition can also be influenced by temperature effects. However, the temperatures needed to see a significant change of the transition crystal fields (without adding charge transfer effects) are so high that the compounds will already decompose. Experimentally, some features have been observed by temperature dependent magnetometry in large magnetic fields. But the effects are too small to be related to a diamagnetic-paramagnetic transition. Therefore, we decided to omit this comment about temperature effects in the revised manuscript.

“Minor correction

1. English has to be polished such as: “electronic structure and ? may become a useful characterization tool for other diamagnetic materials like e.g. the surface region of superconductors or certain topological insulators”. What is ? (I guess magnetic)”

- ▶ The manuscript has been checked again for typos and misspellings.

“2. The abstract needs to be reformulated and made more clear and attractive for the reader. We don’t understand the purpose of the present research when we read it.”

- ▶ The abstract was rewritten and shortened.

“3. Some figures need to be plotted clearly, such as those reported in supplementary information.”

- ▶ All figures included have a resolution of 300 dpi.

REVIEWERS' COMMENTS:

Reviewer #1 (Remarks to the Author):

I have reviewed the paper for the second time. It is a very nice paper, but I am still not convinced about the general impact of the manuscript. I also see that Referee 2 shares, more or less, my opinion about the conclusions. The authors have adjusted the paper to accommodate my comments, but, in the end, that demonstrates that my initial conclusion was correct. Therefore, my recommendation regarding the general impact has not changed.

Reviewer #2 (Remarks to the Author):

The revised manuscript and replies from the authors give all answers for criticisms from reviewers. Although I cannot agree with some answers, especially the separation between Pd M and O K-edge line shapes, it will be discussed in other researchers and will stimulate the community in magnetic spectroscopies.

Following two revisions are necessary.

Since the authors added new figures for DFT calculation, charge density should be discussed. For the analysis of XMCD, the authors used hole number $n_h=2$. As they mentioned the hybridization effect in revised p. 5, estimated charge number from the calculations should be added. The authors should discuss the charge numbers from views points of both DFT and ligand field calculation, which is a main factor for error bars in sum-rule analyses.

In order to reproduce the XMCD experiment by readers, the authors should add more detail information. For XMCD measurement, are powder samples used and mounted in the folder for low temperature measurement, or pellet shape ? How did the authors avoid charge-up effect in TEY measurements in insulators ?

Reviewer #3 (Remarks to the Author):

The authors have replied to my concerns and implemented my suggestions. They have also corrected the manuscript according to the other reviewers questions and suggestions. After reading it carefully, the present version of the manuscript meets the requirements and the high standard of commschem at Nature edition.

We thank all reviewers for their consideration and helpful comments. A detailed point-by-point answer is given below, changes were tracked in the manuscript.

Answer to Reviewer #1:

„I have reviewed the paper for the second time. It is a very nice paper, but I am still not convinced about the general impact of the manuscript. I also see that Referee 2 shares, more or less, my opinion about the conclusions. The authors have adjusted the paper to accommodate my comments, but, in the end, that demonstrates that my initial conclusion was correct. Therefore, my recommendation regarding the general impact has not changed.“

- ▶ We regret that the revised manuscript could not convince the Reviewer from the perspective of possible impact. Since a future impact is usually difficult to predict, we cannot argue against this opinion. From our point of view, the results are very interesting e.g. for the x-ray absorption community, because the soft XMCD obtained from a diamagnet is reported for the first time.

Answer to Reviewer #2:

„The revised manuscript and replies from the authors give all answers for criticisms from reviewers. Although I cannot agree with some answers, especially the separation between Pd M and O K-edge line shapes, it will be discussed in other researchers and will stimulate the community in magnetic spectroscopies. Following two revisions are necessary.

Since the authors added new figures for DFT calculation, charge density should be discussed. For the analysis of XMCD, the authors used hole number $n_h=2$. As they mentioned the hybridization effect in revised p. 5, estimated charge number from the calculations should be added. The authors should discuss the charge numbers from views points of both DFT and ligand field calculation, which is a main factor for error bars in sum-rule analyses.“

- ▶ The Reviewer is right that d holes obtained from DFT should be preferred. We added the calculated numbers of d holes in Tables 2 and 3 and used these values for the sum-rule based analysis of magnetic moments. We obtained $n_h = 1.46-1.47$ for palladium(II) in the shell and $n_h = 1.70$ for the central palladium(II) ion. The numbers of d holes is smaller than 2 because of hybridisation effects, which were already mentioned in the manuscript. As a consequence, all spin and orbital magnetic moments are slightly reduced (proportional to n_h), but the interpretation is not affected.

We added a comment on the dependence of computed numbers of holes on the basis set in the DFT part. Unfortunately, we did not manage to use a (more or less) basis-independent population analysis method, because the use of the effective core potential for Pd caused problems beyond a simple offset that we could not solve in acceptable time. However, since the used numbers of d holes are presented in the same tables as the magnetic moments, they could be easily adjusted for comparison in future work.

„In order to reproduce the XMCD experiment by readers, the authors should add more detail information. For XMCD measurement, are powder samples used and mounted in the folder for low temperature measurement, or pellet shape ? How did the authors avoid charge-up effect in TEY measurements in insulators ?“

- ▶ The sample preparation was mentioned in the Methods section. We re-dispersed the powder samples and drop coated them on freshly cleaved HOPG substrates. In the revised manuscript, we added more details in the Methods section and provide photographs of the sample in the new Supplementary Figure 1.

Charging effects were avoided by the low coverage of the material and the excellent conductivity of the substrate.

Answer to Reviewer #3:

„The authors have replied to my concerns and implemented my suggestions. They have also corrected the manuscript according to the other reviewers questions and suggestions. After reading it carefully, the present version of the manuscript meets the requirements and the high standard of commschem at Nature edition.“

- ▶ We thank the Reviewer for positive evaluation of the revised manuscript.